Corrected: Author correction

# Strong indirect coupling between graphene-based mechanical resonators via a phonon cavity

Gang Luo[1,2], Zhuo-Zhi Zhang [1,2], Guang-Wei Deng [1,2], Hai-Ou Li[1,2], Gang Cao[1,2], Ming Xiao[1,2], Guang-Can Guo[1,2], Lin Tian [3] & Guo-Ping Guo [1,2]

Mechanical resonators are promising systems for storing and manipulating information. To transfer information between mechanical modes, either direct coupling or an interface between these modes is needed. In previous works, strong coupling between different modes in a single mechanical resonator and direct interaction between neighboring mechanical resonators have been demonstrated. However, coupling between distant mechanical resonators, which is a crucial request for long-distance classical and quantum information processing using mechanical devices, remains an experimental challenge. Here, we report the experimental observation of strong indirect coupling between separated mechanical resonators in a graphene-based electromechanical system. The coupling is mediated by a far-off-resonant phonon cavity through virtual excitations via a Raman-like process. By controlling the resonant frequency of the phonon cavity, the indirect coupling can be tuned in a wide range. Our results may lead to the development of gate-controlled all-mechanical devices and open up the possibility of long-distance quantum mechanical experiments.

[1] CAS Key Laboratory of Quantum Information, University of Science and Technology of China, Hefei, 230026 Anhui, China. [2] Synergetic Innovation Center of Quantum Information and Quantum Physics, University of Science and Technology of China, Hefei, 230026 Anhui, China. [3] School of Nature Sciences, University of California, Merced, CA 95343, USA. Gang Luo and Zhuo-Zhi Zhang contributed equally to this work. Correspondence and requests for materials should be addressed to G.-W.D. (email: gwdeng@ustc.edu.cn) or to L.T. (email: ltian@ucmerced.edu) or to G.-P.G. (email: gpguo@ustc.edu.cn)

The rapid development of nanofabrication technology enables the storage and manipulation of phonon states in micro- and nano-mechanical resonators[1–5]. Mechanical resonators with quality factors[6] exceeding 5 million and frequencies[7,8] in the sub-gigahertz range have been reported. These advances have paved the route to controllable mechanical devices with ultralong memory time[9]. To transfer information between different mechanical modes, tunable interactions between these modes are required[10]. While different modes in a single mechanical resonator can be coupled by parametric pump[3,4,11–16] and neighboring mechanical resonators can be coupled via phonon processes through the substrate[2] or direct contact interaction[17], it is challenging to directly couple distant mechanical resonators.

Here, we observe strong effective coupling between mechanical resonators separated at a distance via a phonon cavity that is significantly detuned from these two resonator modes. The coupling is generated via a Raman-like process through virtual excitations in the phonon cavity and is tunable by varying the frequency of the phonon cavity. Typically, a Raman process can be realized in an atom with three energy levels in the Λ form[18,19]. The two lower energy levels are each coupled to the third energy level via an optical field with detunings. When these two detunings are tuned to be equal to each other, an effective coupling is formed between the lower two levels. To our knowledge, tunable indirect coupling in electro-mechanical systems has not been demonstrated before. The physical mechanism of this coupling is analogous to the coupling between distant qubits in circuit quantum electrodynamics[20,21], where the interaction between qubits is induced by virtual photon exchange via a superconducting microwave resonator.

## Results

### Sample characterization

The sample structure is shown in Fig. 1a, where a graphene ribbon[22,23] with a width of ~1 μm and ~5 layers is suspended over three trenches (2 μm in width, 150 nm in depth) between four metal (Ti/Au) electrodes. This configuration defines three distinct electromechanical resonators: $R_1$, $R_2$ and $R_3$. The metallic contacts S and $D_3$ are each 2 μm wide and $D_1$ and $D_2$ are each 1.5 μm wide, which leads to a 7-μm separation between the centers of $R_1$ and $R_3$ (see Supplementary Methods and Supplementary Fig. 1). All measurements are performed in a dilution refrigerator at a base temperature of approximately 10 mK and at pressures below $10^{-7}$ torr. The suspended resonators are biased by a dc gate voltage ($V_{gi}^{DC}$ for the $i$th resonator) and actuated by an ac voltage ($V_{gi}^{AC}$ for the $i$th resonator with driving frequency $f_{gi} = \omega_d/2\pi$) through electrodes ($g_i$ for the $i$th resonator) underneath the respective resonators. To characterize the spectroscopic properties of the resonators, a driving tone is applied to one or more of the bottom gates with frequency $\omega_d$, and another microwave tone with frequency $\omega_d + \delta\omega$ is applied to the contact S. A mixing current ($I_{mix} = I_x + jI_y$) can then be obtained at $D_3$ ($D_1$ and $D_2$ are floated during all measurements) by detecting the δω signal with a lock-in amplifier fixed at zero phase during all measurements (see Supplementary Methods and Supplementary Fig. 2).

Figure 1b shows the measured mixing current as a function of the dc gate voltage and the ac driving frequency on $R_3$, where the oblique lines represent the resonant frequencies of the resonator modes. We denote the resonant frequency of the $i$th resonator as $f_{mi} = \omega_{mi}/2\pi$. This plot shows that $df_{m3}/dV_{g3}^{DC} \sim 7.7$ MHz/V when $|V_{g3}^{DC}| > 5$ V. The frequencies of the resonators can hence be tuned in a wide range (see Supplementary Note 1 and Supplementary Fig. 3 for results of $R_1$ and $R_2$), which allows us to adjust the mechanical modes to be on or off resonance with

each other. The quality factors ($Q$) of the resonant modes are determined by fitting the measured spectral widths (see Supplementary Fig. 8) at low driving powers (typically −50 dBm). Figure 1c shows the spectral dependence of $R_3$, which gives a linewidth of $\gamma_3/2\pi \sim 28$ kHz at a resonant frequency of $f_{m3} \sim 98.05$ MHz. The resulting quality factor is $Q \sim 3500$. The quality factors of the other two resonators are similar, at ~3000.

### Strong coupling between neighboring resonators

Neighboring resonators in this system couple strongly with each other, similar to previous studies on gallium arsenide[2] and carbon nanotube[17]. Figure 1d, e shows the spectra of the coupled modes ($R_1$, $R_2$) and ($R_2$, $R_3$), respectively, by plotting the mixed current $I_x$ as a function of gate voltages and driving frequencies. In Fig. 1e, the voltage $V_{g3}^{DC}$ is fixed at 10.5 V, with a corresponding resonant frequency $f_{m3} = 101.15$ MHz, and $V_{g2}^{DC}$ is scanned over a range with $f_{m2}$ being near-resonant to $f_{m3}$. A distinct avoided level crossing appears when $f_{m2}$ approaches $f_{m3}$, which is a central feature of two resonators with direct coupling. From the measured data, we extract the coupling rate between these two modes as $\Omega_{23}/2\pi \sim 200$ kHz, which is the energy splitting when $f_{m2} = f_{m3}$. In Supplementary Note 2 and Supplementary Fig. 4, we fit the measured spectrum with a single two-mode model using this coupling rate. Similarly in Fig. 1d, by fixing $V_{g1}^{DC}$ at 10.5 V and scanning the voltage $V_{g2}^{DC}$, we obtain the coupling rate between $R_1$ and $R_2$ as $\Omega_{12}/2\pi \sim 240$ kHz. There are several possible origins for the coupling between two adjacent resonators in this system. One coupling medium is the substrate and the other medium is the graphene ribbon itself. Mechanical energy can be transferred in a solid-state material by phonon propagation, as demonstrated in several experiments[2,17,24]. Second, because adjacent resonators share lattice bonds, the phonon energy can transfer in the graphene ribbon. The dependence of the coupling strength on the width of the drain contacts is still unknown (see Supplementary Fig. 5 for another sample).

The measured coupling strength satisfies the strong coupling condition with $\Omega_{23} \gg \gamma_2, \gamma_3$. Defining the cooperativity for this phonon–phonon coupling system as $C = \Omega_{23}^2/\gamma_2\gamma_3$, we find that $C = 44$. A similar strong coupling condition can be found between modes $R_1$ and $R_2$. By adjusting the gate voltages of these three resonators, $R_2$ can be successively coupled to both $R_1$ and $R_3$ (see Fig. 1f).

For comparison, we study the coupling strength between modes $R_1$ and $R_3$. The frequency $f_{m2}$ of resonator $R_2$ is set to be detuned from $f_{m1}$ and $f_{m3}$ by 700 kHz in Fig. 2a. In the dashed circle, we observe a near-perfect level crossing when $f_{m1}$ approaches $f_{m3}$, which indicates a negligible coupling between these two modes, with $\Omega_{13} \ll \gamma_1, \gamma_3$ (also see Supplementary Fig. 6).

### Raman-like coupling between well-separated resonators

The three resonator modes in our system are in the classical regime. The Hamiltonian of these three classical resonators can be written as:

$$\mathcal{H}_c = \sum_i^3 \frac{1}{2}(p_i^2 + \omega_{mi}^2 x_i^2) + \Lambda_{12}x_1x_2 + \Lambda_{23}x_2x_3, \quad (1)$$

where $\Lambda_{ij} = \Omega_{ij}\sqrt{\omega_{mi}\omega_{mj}}$ is a coupling parameter between $i$- and $j$th resonators, $p_{pi}$ is the effective momentum and $x_i$ is the effective coordinate of the oscillation for the $i$th resonator, respectively. Let $x_i = \sqrt{\frac{1}{2\omega_{mi}}}(\alpha_i^* + \alpha_i)$ and $p_i = i\sqrt{\frac{\omega_{mi}}{2}}(\alpha_i^* - \alpha_i)$, with $\alpha_i$ and $\alpha_i^*$ being complex numbers. The Hamiltonian in Eq.

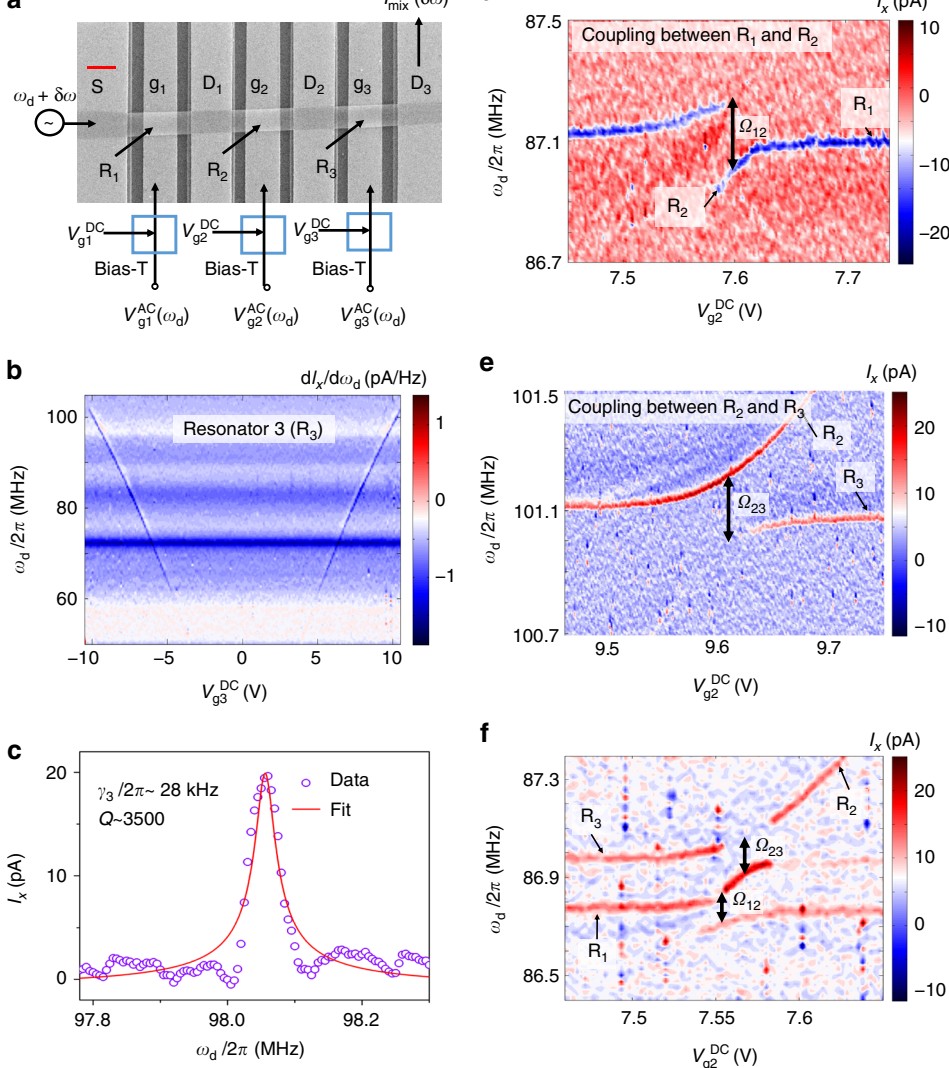

**Fig. 1** Sample structure and device characterization. **a** Scanning electron microscope photograph of a typical sample. An ~1-μm-wide graphene ribbon was suspended over four contacts, labeled as S, $D_1$, $D_2$, and $D_3$, respectively. These contacts divide the ribbon into three sections, each with a gate of ~150 nm beneath the ribbon. A driving microwave with frequency $\omega_d + \delta\omega$ is applied to contact S and is detected at contact $D_3$ after mixing with another driving tone with frequency $\omega_d$ applied to one or more of the control gates. Scale bar is 1 μm. **b** The differentiation of the mixed current $dI_x/d\omega_d$ as a function of driving frequency $\omega_d$ and gate voltage $V_{g3}^{DC}$ with $V_{g1}^{DC} = V_{g2}^{DC} = 0$ V. Here, the frequencies of all resonators can be tuned from several tens of MHz to ~100 MHz by adjusting the dc gate voltages. **c** The mixing current as a function of the driving frequency $\omega_d$ at voltage $V_{g3}^{DC} = 10$ V. Using a fitting process (see Supplementary Fig. 8), we extract the linewidth of the mechanical mode. The data were obtained at a driving power of −5 dBm. **d**, **e** Spectra of coupled modes $R_1$ and $R_2$ (**d**, where $V_{g1}^{DC} = 10.5$ V and $V_{g3}^{DC} = 0$ V) and $R_2$ and $R_3$ (**e**, where $V_{g1}^{DC} = 0$ V and $V_{g3}^{DC} = 10.5$ V). Strong couplings between these modes are manifested as avoided level crossings in the plots. Coupling strengths $\Omega_{12}/2\pi \sim 240$ kHz and $\Omega_{23}/2\pi \sim 200$ kHz are extracted from the plots. **f** The spectrum of $R_2$ coupled to both $R_1$ and $R_3$. In this case, the gate voltages $V_{g1}^{DC} = 10.45$ V and $V_{g3}^{DC} = 8.35$ V are fixed

(1) can be written as

$$\mathcal{H}_t = \sum \omega_{mi}\alpha_i^*\alpha_i + \frac{\Omega_{12}}{2}\left(\alpha_1^*\alpha_2 + \alpha_1\alpha_2^*\right) + \frac{\Omega_{23}}{2}\left(\alpha_2^*\alpha_3 + \alpha_2\alpha_3^*\right). \quad (2)$$

Here, we have applied the rotating-wave approximation and neglected the $\alpha_i\alpha_j$ and $\alpha_i^*\alpha_j^*$ terms. This approximation is valid when $\omega_{mi} \gg \Omega_{12}, \Omega_{23}$. This Hamiltonian describes the direct couplings between neighboring resonators ($R_1$, $R_2$) and ($R_2$, $R_3$). Through these couplings, the mechanical modes hybridize into three normal modes, and an effective coupling between modes $R_1$ and $R_3$ can be achieved. If the resonators work in the quantum

regime, $\alpha_i$ and $\alpha_i^*$ can be quantized into the annihilation and creation operators of a quantum harmonic oscillator, respectively.

We study the hybridization of this three-mode system by fixing the gate voltages (mode frequencies) of modes $R_1$ and $R_2$, and sweeping the gate voltage of $R_3$ over a wide range. The spectrum of this system depends strongly on the detuning between modes $R_1$ and $R_2$, which is defined as $\Delta_{12} = 2\pi(f_{m2}-f_{m1})$. In Fig. 2a, $\Delta_{12}/2\pi \sim 70$ kHz. Similar to Fig. 1f, modes $R_3$ and $R_1$ show a level crossing. Moreover, we observe a large avoided level crossing between modes $R_2$ and $R_3$ when the frequency $f_{m3}$ approaches $f_{m2}$, indicating strong coupling between these two modes. Hence, even with strong couplings between all neighboring resonators, the effective coupling between the distant modes $R_1$ and $R_3$ is still negligible when the frequency of mode $R_2$ is significantly far off

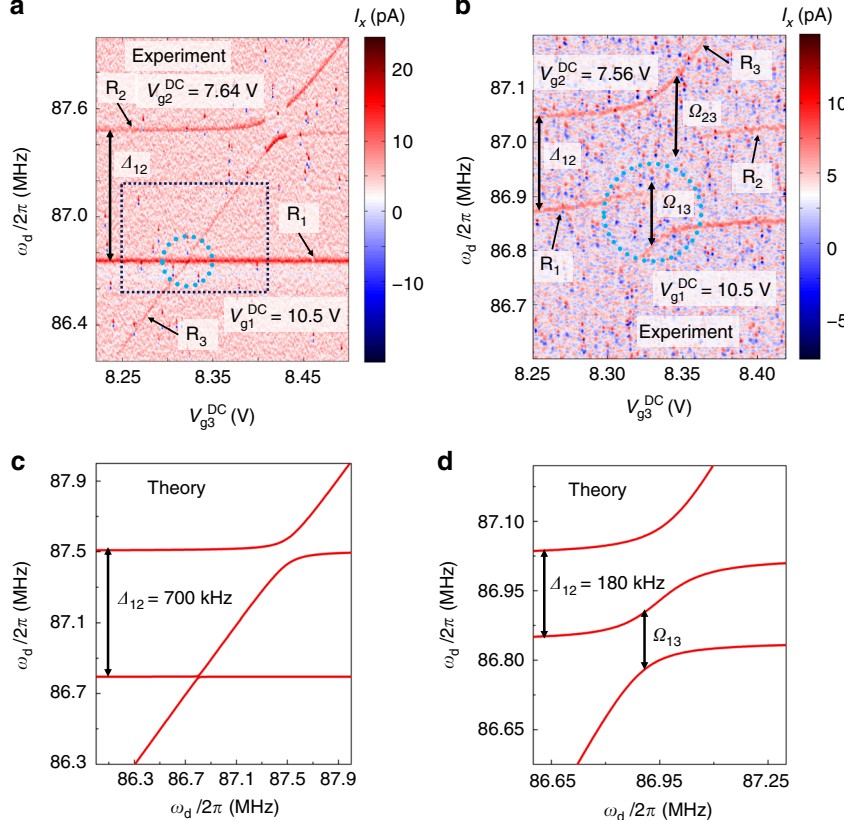

**Fig. 2** Hybridization between all three modes. **a** Measured spectrum of the three-mode system when the frequency of R$_2$ is far off-resonance from that of mode R$_1$ by a detuning $\Delta_{12}/2\pi \sim$ 70 kHz (here, $V_{g1}^{DC}$ = 10.5 V and $V_{g2}^{DC}$ = 7.64 V). The dc voltage $V_{g3}^{DC}$ is scanned over a wide range, crossing both $f_{m1}$ and $f_{m2}$. An avoided level crossing is observed when $f_{m3}$ approaches $f_{m2}$. A level crossing is observed when $f_{m3}$ approaches $f_{m1}$. **b** Measured spectrum of the three-mode system when the detuning is $\Delta_{12}/2\pi \sim$ 180 kHz (here, $V_{g1}^{DC}$ = 10.5 V and $V_{g2}^{DC}$ = 7.56 V, and here the ranges of the axes are set to be the same as the black dashed box shown in **a**). Here, a strongly avoided level crossing appears when $f_{m3}$ approaches $f_{m1}$. The strengths of the direct couplings extracted from the measured spectrum are $\Omega_{12}/2\pi$ = 240 kHz and $\Omega_{23}/2\pi$ = 170 kHz. **c, d** Spectra calculated using the theoretical model for the three modes (Eq. (2)) and coupling constants $\Omega_{12}$ and $\Omega_{23}$. $\Delta_{12}/2\pi$ = 700 kHz in **c** and $\Delta_{12}/2\pi$ = 180 kHz in **d**

resonance from the other two modes. On the contrary, when the detuning $\Delta_{12}/2\pi$ is lowered to ~180 kHz, a distinct avoided level crossing between modes R$_1$ and R$_3$ is observed, as shown inside the dashed circle in Fig. 2b.

With coupling strengths $\Omega_{12}/2\pi$ = 240 kHz and $\Omega_{23}/2\pi$ = 170 kHz extracted from the measured data, we plot the theoretical spectra of the normal modes in this three-mode system given by Eq. (2), for $\Delta_{12}/2\pi$ = 700 and 180 kHz in Fig. 2c, d, respectively. Our result shows good agreement between theoretical and experimental results.

With direct couplings between neighboring resonators, an effective coupling between the two distant resonators R$_1$ and R$_3$ can be obtained via their couplings to mode R$_2$. The effective coupling can be viewed as a Raman process, as illustrated in Fig. 3a. Here mode R$_2$ functions as a phonon cavity that connects the mechanical resonators R$_1$ and R$_3$ via virtual phonon excitations. The physical mechanism of this effective coupling is similar to that of the coupling between distant superconducting qubits via a superconducting microwave cavity[20]. The detuning between the phonon cavity and the other two modes $\Delta_{12}$ can be used as a control parameter to adjust this effective coupling.

To derive the effective coupling, we consider the case of $\Delta_{12} = \Delta_{32} = \Delta$, where $\Delta_{32}/2\pi = f_{m2} - f_{m3}$ and $|\Delta| \gg \Omega_{12}, \Omega_{23}$. The avoided level crossing between modes R$_1$ and R$_3$ can be extracted at this point. Using a perturbation theory approach, we obtain the

effective Hamiltonian between modes R$_1$ and R$_3$ as (see Methods for details)

$$\mathcal{H}_{\text{eff}} = \left(\Delta + \frac{\Omega_{12}^2}{4\Delta}\right)\alpha_1^*\alpha_1 + \left(\Delta + \frac{\Omega_{23}^2}{4\Delta}\right)\alpha_3^*\alpha_3 + \frac{\Omega_{13}}{2}\left(\alpha_1^*\alpha_3 + \alpha_3^*\alpha_1\right).$$

$$(3)$$

Here, an effective coupling is generated between R$_1$ and R$_3$ with magnitude $\Omega_{13} = \Omega_{12}\Omega_{23}/2\Delta$, and the resonant frequencies of each mode are shifted by a small term. The effective coupling $\Omega_{13}$ in the Hamiltonian depends strongly on the detuning $\Delta$. Thus, the effective coupling between R$_1$ and R$_3$ can be controlled over a wide range by varying the frequency (gate voltage) of resonator R$_2$.

The effective coupling strength $\Omega_{13}$ between R$_1$ and R$_3$ as a function of $\Delta_{12}$ is shown in Fig. 3b. Each data point is obtained by changing the gate voltage of R$_2$ and repeating the measurements in Fig. 2a, b (see Supplementary Fig. 7). Over a large range of detuning, the effective coupling is larger than the linewidths of the resonators $\gamma_{1,2,3}/2\pi$, with $\Omega_{13} >$ 30 kHz. The red line shows the results using perturbation theory. The experimental data indicate good agreement with the theoretical results.

## Discussion

In summary, we have demonstrated indirect coupling between separated mechanical resonators in a three-mode electromechanical

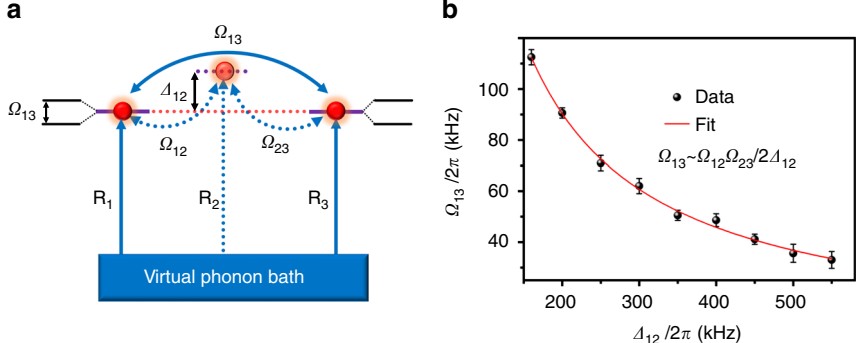

**Fig. 3** Indirect coupling between separated resonators via a phonon cavity. **a** Raman-like coupling between modes $R_1$ and $R_3$ via virtual excitation of the phonon cavity $R_2$. The coupling strength can be controlled by changing the detuning $\Delta_{12}$. **b** Effective coupling as a function of $\Delta_{12}$. The error bars are obtained from the s. e. m. of the measured data and are extracted from the statistical deviation of the estimated values at different detunings from Supplementary Fig. 7. The red line is given by $\Omega_{13} = \Omega_{12}\Omega_{23}/2\Delta_{12}$, with $\Omega_{12}/2\pi = 240$ kHz and $\Omega_{23}/2\pi = 170$ kHz

system constructed from a graphene ribbon. Our study suggests that coupling between well-separated mechanical modes can be created and manipulated via a phonon cavity. These observations hold promise for a wide range of applications in phonon state storage, transmission, and transformation. In the current experiment, the sample works in an environment subjected to noise and microwave heating with typical temperatures as high as 100 mK and phonon numbers reaching ~24. By cooling the mechanical resonators to lower temperatures[25–28], quantum states could be manipulated via this indirect coupling[29,30]. Furthermore, in the quantum limit, by coupling the mechanical modes to solid-state qubits, such as quantum-dots and superconducting qubits[17,31,32], this system can be utilized as a quantum data bus to transfer information between qubits[33,34]. Future work may lead to the development of graphene-based mechanical resonator arrays as phononic waveguides[24] and quantum memories[35] with high tunabilities.

## Methods

**Theory of three-mode coupling.** We describe this three-mode system with the Hamiltonian ($\hbar = 1$)

$$\mathcal{H}_t = \sum \omega_{mi}\alpha_i^*\alpha_i + \frac{\Omega_{12}}{2}(\alpha_1^*\alpha_2 + \alpha_1\alpha_2^*) + \frac{\Omega_{23}}{2}(\alpha_2^*\alpha_3 + \alpha_2\alpha_3^*), \quad (4)$$

where $\Omega_{ij}$ is the coupling between mechanical resonators $i$ and $j$. The couplings between the resonators induce hybridization of the three modes. The hybridized normal modes under this Hamiltonian can be obtained by solving the eigenvalues of the matrix

$$M = \begin{pmatrix} \Delta_{12} & \frac{\Omega_{12}}{2} & 0 \\ \frac{\Omega_{12}}{2} & 0 & \frac{\Omega_{23}}{2} \\ 0 & \frac{\Omega_{23}}{2} & \Delta_{23} \end{pmatrix}, \quad (5)$$

where $\Delta_{ij}/2\pi = f_{mi} - f_{mj}$ is the frequency difference between $R_i$ and $R_j$. The eigenvalues of this matrix correspond to the frequencies of the normal modes, i.e., the peaks in the spectroscopic measurement.

We consider the special case of $\Delta_{12} = \Delta_{23} = \Delta$, with $|\Delta| \gg \Omega_{12}, \Omega_{23}$, in the three-mode system. Here, the eigenvalues of the normal modes can be derived analytically. One eigenvalue is $\omega_\Delta = \Delta$, which corresponds to the eigenmode

$$\alpha_\Delta = -\frac{\Omega_{23}}{\sqrt{\Omega_{12}^2 + \Omega_{23}^2}}\alpha_1 + \frac{\Omega_{12}}{\sqrt{\Omega_{12}^2 + \Omega_{23}^2}}\alpha_3. \quad (6)$$

This mode is a superposition of the end modes $\alpha_1$ and $\alpha_3$, and does not include the middle mode. The two other eigenvalues are

$$\omega_{\Delta\pm} = \frac{1}{2}(\Delta \pm \omega_{\Delta 0}) \quad (7)$$

with $\omega_{\Delta 0} = \sqrt{\Delta^2 + \Omega_{12}^2 + \Omega_{23}^2}$. The corresponding normal modes are

$$a_{\Delta\pm} = \frac{(\Omega_{12}a_1 \pm (\omega_{\Delta 0} \mp \Delta)a_2 + \Omega_{23}a_3)}{\sqrt{2\omega_{\Delta 0}(\omega_{\Delta 0} \mp \Delta)}}. \quad (8)$$

With $|\Delta| \gg \Omega_{12}, \Omega_{23}$, for $\Delta > 0$, $\omega_{\Delta+} \approx \Delta + (\Omega_{12}^2 + \Omega_{23}^2)/4\Delta$. The mode $\alpha_{\Delta+}$ is nearly degenerate with $\alpha_\Delta$, and

$$\alpha_{\Delta+} \approx \frac{\Omega_{12}\alpha_1 + \Omega_{23}\alpha_3}{\sqrt{\Omega_{12}^2 + \Omega_{23}^2}}. \quad (9)$$

The mode $\alpha_{\Delta-}$ has frequency $\omega_{\Delta-} \approx -(\Omega_{12}^2 + \Omega_{23}^2)/4\Delta$, with $\alpha_{\Delta-} \approx \alpha_2$. The normal modes now become separated into two nearly degenerate modes $\{\alpha_\Delta, \alpha_{\Delta+}\}$, which are superpositions of modes $\alpha_1$ and $\alpha_3$, and a third mode $\alpha_{\Delta-}$ that is significantly off resonance from the other two modes. The nearly degenerate modes can be viewed as a hybridization of $\alpha_1$ and $\alpha_3$ with an effective splitting $(\Omega_{12}^2 + \Omega_{23}^2)/2\Delta$. A similar result can be derived for $\Delta < 0$, where $\omega_{\Delta-} \approx \Delta + (\Omega_{12}^2 + \Omega_{23}^2)/4\Delta$, with $\alpha_{\Delta-}$ given by the expression in Eq. (8), and $\omega_{\Delta+} \approx -(\Omega_{12}^2 + \Omega_{23}^2)/4\Delta$ with $\alpha_{\Delta+} \approx \alpha_2$.

The effective coupling rate can be derived with a perturbative approach on the matrix $M$. When $|\Delta| \gg \Omega_{12}, \Omega_{23}$, the dynamics of $\alpha_1$ and $\alpha_3$ is governed by matrix

$$M_{eff} = \begin{pmatrix} \Delta + \frac{\Omega_{12}^2}{4\Delta} & \frac{\Omega_{12}\Omega_{23}}{4\Delta} \\ \frac{\Omega_{12}\Omega_{23}}{4\Delta} & \Delta + \frac{\Omega_{23}^2}{4\Delta} \end{pmatrix}. \quad (10)$$

This matrix tells us that because of their interaction with the middle mode $\alpha_2$, the frequency of mode $\alpha_1$ ($\alpha_3$) is shifted by $\frac{\Omega_{12}^2}{4\Delta}$ ($\frac{\Omega_{23}^2}{4\Delta}$), which is much smaller than $|\Delta|$. Meanwhile, an effective coupling is generated between these two modes with magnitude $\Omega_{13} = \frac{\Omega_{12}\Omega_{23}}{2\Delta}$. The effective Hamiltonian for $\alpha_1$ and $\alpha_3$ can be written as

$$\mathcal{H}_{eff} = (\Delta + \frac{\Omega_{12}^2}{4\Delta})\alpha_1^*\alpha_1 + (\Delta + \frac{\Omega_{23}^2}{4\Delta})\alpha_3^*\alpha_3 + \frac{\Omega_{13}}{2}(\alpha_1^*\alpha_3 + \alpha_3^*\alpha_1). \quad (11)$$

The effective coupling can be controlled over a wide range by varying the frequency of the second mode $\alpha_2$.

**Data availability.** The remaining data contained within the paper and Supplementary files are available from the author upon request.

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

## Acknowledgements

This work was supported by the National Key R&D Program of China (Grant No. 2016YFA0301700), the NSFC (Grants Nos. 11625419, 61704164, 61674132, 11674300, 11575172, and 91421303), the SPRP of CAS (Grant No. XDB01030000), and the Fundamental Research Fund for the Central Universities. L.T. is supported by the National Science Foundation under Award No. DMR-0956064 and PHY-1720501 and the UC Multicampus-National Lab Collaborative Research and Training under Award No. LFR-17-477237. This work was partially carried out at the USTC Center for Micro and Nanoscale Research and Fabrication.

## Author contributions

G.L. and Z.-Z.Z. fabricated the device. G.-W.D. and Z.-Z.Z. performed the measurements. L.T., G.-W.D, and Z.-Z.Z analyzed the data and developed the theoretical analysis. H.-O.L, G.C., M.X., and G.-C.G. supported the fabrication and measurement. G.-P.G. and G.-W.D. planned the project. All authors participated in writing the manuscript.

## Additional information

**Competing interests:** The authors declare no competing financial interests.

