## [Peer Review File · Nature Communications]

Reviewers' comments:

Reviewer #1 (Remarks to the Author):

As the reviewer 1 for the previous submission of this manuscript to Nature Nanotechnology, I am mostly satisfied with the responses from the authors—most questions and concerns have been appropriately addressed, and the manuscript has been improved. I will recommend publication of this revised manuscript in Nature Communications once the few remaining issues are properly addressed.

The writing should be more objective. For example, when describing the current Fig. 1f, in the caption the writing is “The resonant frequency of R_2 ($V_{g2DC}=7.64$ V) is set to be significantly detuned from that of R_1 and R_3 ”. Here, “significantly” is not an objective description without quantifying the actual detuning value. Readers may look up the value of f_{m2} at this gate voltage in Fig. S3b and find that at $V_{g2DC}=7.64$ V, f_{m2} is quite close to where f_{m1} and f_{m3} crosses (actually, within 1 MHz), especially when compared with the data range shown in Fig. 1b, S3a, S3b. Therefore, whether the detuning is significant or not depends on the context, and the discussion should be put in perspective. A more objective description, like the one used in the text “The frequency f_{m2} of resonator R_2 is detuned from f_{m1} and f_{m3} by 700kHz”, or as in Fig. 2a caption, would be more appropriate.

Regarding previous question 9 (on Fig. 2, the most important result in this work) the authors have made improvements, but not quiet to the expected level. For example, a dashed box in Fig. 2a was added, but was NOT mentioned at all in either caption or text (at least I was not able to find any reference to that box), which will confuse the readers—especially that it happens to also outline the sub-dataset in Fig. 1f (except the gate voltage range is slightly off). Also the contrast between Fig. 2a and 2b is still not clear enough, as they have quite different vertical ranges.

Therefore I make the following recommendations. If possible, the authors should consider:

1. Moving current Fig. 1f to Fig. 2. So in Fig. 2 there will be both zoomed-out (current Fig. 2a) and zoomed-in (current Fig. 1f) data of the non-coupling case between R_1 and R_3 .
2. Show zoomed-out data for the coupling case. The current Fig. 2b shows the zoomed-in data of the coupling case (and makes a good side-to-side comparison with current Fig. 1f, which is suggested to be moved to Fig. 2), but a zoom-out data plot would make a side-to-side comparison with current Fig. 2a and be very helpful for the readers to visualize the case at a larger scale.
3. Replace the current Fig. 1f (which is suggested to be moved to Fig. 2) with a large range plot showing R_2 couples to both R_1 and R_3 (to some extent similar to the current Fig. S5).

This way, the entire Fig. 1 describes the coupling between adjacent resonators, while Fig. 2 entirely focus on remote coupling between R_1 and R_3 , and how such coupling can be tuned through R_2 . This would be a more clear-cut story telling.

Similarly, in current Fig. S6 only the zoomed-in data is shown. If the authors can also show the zoomed-out data (showing all the three resonances) for each panel it would be very helpful, as the readers can clearly see how the anti-crossing widens as f_{m2} approaches f_{m1} and f_{m3} .

Besides these suggested modifications, a few glitches (likely due to carelessness) also need to be fixed. For example, in response to my original point 10, the authors wrote “We have also added a vertical dashed line to illustrate that the original frequency of R_1 equals to that of R_3 ”, while in fact they have added a horizontal dashed line. Similarly, in the caption of current Fig. S5, at least one of the V_{g1DC} is mislabeled: “Here $V_{g1DC} = 13$ V and $V_{g1DC} = 9:6$ V.”

I also urge the authors to carefully check the labels inside Fig. S5. Here, assuming all the labels (R_1 , R_2 , R_3) are correct, one will read from the dispersion relationship that $f_{m3}=101$ MHz roughly at $V_{g3DC}=9.6$ V. However, from both Fig. 1b and the caption of Fig. 1e one can find that $f_{m3}=101$ MHz roughly at $V_{g3DC}=10.5$ V. In contrast, from Fig. S3 and Fig. 1e one can estimate that that $f_{m2}=101$ MHz roughly at $V_{g2DC}=9.6$ V. Therefore the authors should carefully check both the label and the caption in the current Fig. S5 to make sure the data presented there (and the information inferred from it) is consistent with all the other plots.

Overall, the manuscript is clearly improved and most questions/concerns have been properly addressed. Upon properly addressing these few remaining issues I will be happy to recommend publication of this manuscript in Nature Communications.

Reviewer #2 (Remarks to the Author):

The revised manuscript clarifies a lot of questions regarding the details that were raised in the previous round of review at Nature Nanotechnology. I appreciate the additional data that the authors have added. The aspects that are still problematic for me

1. Mislabelling of data

(a) In fig.1b caption they mention the plot to be of I_{mix} , whereas in the plot the axis is written as derivative of I_x .

(b) In page 5/13 they mention of plotting $d(I_{\text{mix}})/d(\omega_d)$. But in corresponding plots, they mention the axis as I_x .

2. The author seem to say that using a Hamiltonian is not wrong. I agree that use of a Hamiltonian based description of the system is not wrong. However, when the system is not quantum mechanical – it is a clearly classical as the authors also point out in their response – use of a toy model that seems to imply that it is quantum mechanical is needless and confusing to readers. When a simpler explanation is available for the observed experiments then why not use a simpler explanation. I urge the authors to provide a classical description that explains their observations.

3. I still think that the authors are still excessively using dashed/dotted lines to indicate trends in their data. The data should speak for itself. Figure 2b is an example of this even in the revised manuscript.

Reviewer #3 (Remarks to the Author):

This paper reports on the experimental demonstration of a tunable, Raman-type coupling of two separated mechanical modes in a graphene-based electromechanical system. As already mentioned in my previous report, I consider the paper very-well written and all the claims and conclusions are convincingly supported by experimental data and a theoretical model. In the resubmitted version the author have addressed all the points raised by the other referees, which further improved the quality of the manuscript. Although I still do not consider the demonstrated coupling as a major breakthrough, it is of relevance in the field of electromechanical systems and potentially in the future for the control of (quantum) mechanical transducers. Therefore, I recommend the paper for publication in Nature Communications.

Reviewers' comments:

Reviewer #1 (Remarks to the Author):

As the reviewer 1 for the previous submission of this manuscript to Nature Nanotechnology, I am mostly satisfied with the responses from the authors—most questions and concerns have been appropriately addressed, and the manuscript has been improved. I will recommend publication of this revised manuscript in Nature Communications once the few remaining issues are properly addressed.

Reply:

We thank Reviewer #1 very much for his/her positive comments and his/her very careful reading of our manuscript. Below we will address all his/her questions/suggestions. We have also revised the manuscript and the supplementary materials thoroughly based on these suggestions.

The writing should be more objective. For example, when describing the current Fig. 1f, in the caption the writing is “The resonant frequency of $V_{g2}^{DC}=7.64$ V) is set to be significantly detuned from that of R_1 and R_3 ”. Here, “significantly” is not an objective description without quantifying the actual detuning value. Readers may look up the value of fm2 at this gate voltage in Fig. S3b and find that at $V_{g2}^{DC}=7.64$ V, fm2 is quite close to where fm1 and fm3 crosses (actually, within 1 MHz), especially when compared with the data range shown in Fig. 1b, S3a, S3b. Therefore, whether the detuning is significant or not depends on the context, and the discussion should be put in perspective. A more objective description, like the one used in the text “The frequency fm2 of resonator R_2 is detuned from fm1 and fm3 by 700kHz”, or as in Fig. 2a caption, would be more appropriate.

Reply:

Thanks for the valuable suggestion. According to this comment and the reviewer’s next comment, we have replaced Fig. 1f by a plot with R2 coupled to both R1 and R3. In addition, all gate voltages used to produce this figure have been mentioned in the caption, and we have also revised the corresponding description in the main text.

Regarding previous question 9 (on Fig. 2, the most important result in this work) the authors have made improvements, but not quiet to the expected level. For example, a dashed box in Fig. 2a was added, but was NOT mentioned at all in either caption or text (at least I was not able to find any reference to that box), which will confuse the

readers—especially that it happens to also outline the sub-dataset in Fig. 1f (except the gate voltage range is slightly off).

Reply:

We sincerely thank the reviewer's comments. We have added a description of the dashed box in the caption of Fig. 2b: **here the axis ranges are set to be the same as the black dashed box shown in panel a.**

Also the contrast between Fig. 2a and 2b is still not clear enough, as they have quite different vertical ranges. Therefore I make the following recommendations. If possible, the authors should consider:

1. Moving current Fig. 1f to Fig. 2. So in Fig. 2 there will be both zoomed-out (current Fig. 2a) and zoomed-in (current Fig. 1f) data of the non-coupling case between R1 and R3.
2. Show zoomed-out data for the coupling case. The current Fig. 2b shows the zoomed-in data of the coupling case (and makes a good side-to-side comparison with current Fig. 1f, which is suggested to be moved to Fig. 2), but a zoom-out data plot would make a side-to-side comparison with current Fig. 2a and be very helpful for the readers to visualize the case at a larger scale.
3. Replace the current Fig. 1f (which is suggested to be moved to Fig. 2) with a large range plot showing R2 couples to both R1 and R3 (to some extent similar to the current Fig. S5).

This way, the entire Fig. 1 describes the coupling between adjacent resonators, while Fig. 2 entirely focus on remote coupling between R1 and R3, and how such coupling can be tuned through R2. This would be a more clear-cut story telling.

Similarly, in current Fig. S6 only the zoomed-in data is shown. If the authors can also show the zoomed-out data (showing all the three resonances) for each panel it would be very helpful, as the readers can clearly see how the anti-crossing widens as f_{m2} approaches f_{m1} and f_{m3} .

Reply:

We sincerely thank the reviewer's detail suggestions. The reviewer suggests that we can show both zoomed-in and zoomed-out data for both the strong coupling and the weak coupling cases. Because of the experimental procedure we used, we only have zoomed-out data for the weak coupling case (the current Fig. 2a). In the experiment, after large-range scanning (to clarify the gate voltages and the corresponding resonant frequencies of all three resonators), we only focus on smaller axis ranges, followed by

data collection at different detunings (Δ_{12}) to produce the current Fig. 2b and Fig. S6.

In the revised manuscript, we add the information about the gate voltages in Fig. 2a and Fig. 2b for the readers to easily see the difference between these two figures. We hope this will make the message clearer to the readers.

Following the third suggestion of the reviewer, we have replaced Fig. 1f by a large plot with R2 coupled to both R1 and R3.

For Fig. S6, as explained above, we only have data in the range of the current plot. But thanks for the suggestions.

Besides these suggested modifications, a few glitches (likely due to carelessness) also need to be fixed. For example, in response to my original point 10, the authors wrote “We have also added a vertical dashed line to illustrate that the original frequency of R1 equals to that of R3”, while in fact they have added a horizontal dashed line. Similarly, in the caption of current Fig. S5, at least one of the V_{g1DC} is mislabeled: “Here $V_{g1DC} = 13 \text{ V}$ and $V_{g1DC} = 9.6 \text{ V}$.”

I also urge the authors to carefully check the labels inside Fig. S5. Here, assuming all the labels (R1, R2, R3) are correct, one will read from the dispersion relationship that $f_{m3}=101\text{MHz}$ roughly at $V_{g3DC}=9.6\text{V}$. However, from both Fig. 1b and the caption of Fig. 1e one can find that $f_{m3}=101\text{MHz}$ roughly at $V_{g3DC}=10.5\text{V}$. In contrast, from Fig. S3 and Fig. 1e one can estimate that that $f_{m2}=101\text{MHz}$ roughly at $V_{g2DC}=9.6\text{V}$. Therefore the authors should carefully check both the label and the caption in the current Fig. S5 to make sure the data presented there (and the information inferred from it) is consistent with all the other plots.

Reply:

Thanks for these comments. The word “vertical” should be replaced by “horizontal”. We are sorry for the confusion in the label of Fig. S5 in previous version (the current Fig. S6). We have also checked the raw data, which give $V_{g1}^{DC} = 13 \text{ V}$ and $V_{g2}^{DC} = 9.7 \text{ V}$. We have added these values to the caption of the corresponding supplementary figure.

Overall, the manuscript is clearly improved and most questions/concerns have been properly addressed. Upon properly addressing these few remaining issues I will be happy to recommend publication of this manuscript in Nature Communications.

Reply:

Thanks very much for the comments/suggestions. We have revised our manuscript following the reviewer's comments/suggestions, which has greatly improved the quality of the paper.

Reviewer #2 (Remarks to the Author):

The revised manuscript clarifies a lot of questions regarding the details that were raised in the previous round of review at Nature Nanotechnology. I appreciate the additional data that the authors have added. The aspects that are still problematic for me

1. Mislabeling of data

(a) In fig.1b caption they mention the plot to be of I_{mix} , whereas in the plot the axis is written as derivative of I_x .

(b) In page 5/13 they mention of plotting $d(I_{\text{mix}})/d(\omega_d)$. But in corresponding plots, they mention the axis as I_x .

Reply:

Thanks for the positive comments. We have corrected the mislabeling:

(a) In Fig. 1b, we revised the caption to be: “**b The differentiation of the mixed current $dI_x/d\omega_d$** as a function of driving frequency ω_d and gate voltage V_{g3}^{DC}

with $V_{g1}^{DC} = V_{g2}^{DC} = 0$ V.”

(b) On page 5/13 (for previous version and 4/13 for revised version), we revised the text to be: “Figure 1d,e show the spectra of the coupled modes (R_1 , R_2) and (R_2 , R_3), respectively, by plotting **the mixed current I_x as a function of gate voltages and driving frequencies.**”

2. The author seem to say that using a Hamiltonian is not wrong. I agree that use of a Hamiltonian based description of the system is not wrong. However, when the system is not quantum mechanical – it is a clearly classical as the authors also point out in their response – use of a toy model that seems to imply that it is quantum mechanical is needless and confusing to readers. When a simpler explanation is available for the

observed experiments then why not use a simpler explanation. I urge the authors to provide a classical description that explains their observations.

Reply:

Thanks for the suggestion. We have followed the reviewer's suggestion and modified the theory. The theory in the main text is now for classical oscillator modes:

The three resonator modes in our system are in the classical regime. The Hamiltonian of these three classical resonators can be written as:

$$\mathcal{H}_c = \sum_i^3 \frac{1}{2} (p_i^2 + \omega_{mi}^2 x_i^2) + \Lambda_{12} x_1 x_2 + \Lambda_{23} x_2 x_3, \quad (1)$$

where $\Lambda_{ij} = \Omega_{ij} \sqrt{\omega_{mi} \omega_{mj}}$ is a coupling parameter between i - and j th resonators, p_i is the effective momentum and x_i is the effective coordinate of the oscillation for the i th resonator, respectively. Let $x_i = \sqrt{\frac{1}{2\omega_{mi}}} (\alpha_i^* + \alpha_i)$ and $p_i = i \sqrt{\frac{\omega_{mi}}{2}} (\alpha_i^* - \alpha_i)$, with α_i and α_i^* being complex numbers, the Hamiltonian in Eq. (1) can be written as

$$\mathcal{H}_t = \sum \omega_{mi} \alpha_i^* \alpha_i + \frac{\Omega_{12}}{2} (\alpha_1^* \alpha_2 + \alpha_1 \alpha_2^*) + \frac{\Omega_{23}}{2} (\alpha_2^* \alpha_3 + \alpha_2 \alpha_3^*). \quad (2)$$

Here we have applied the rotating-wave approximation and neglected the $\alpha_i \alpha_j$ and $\alpha_i^* \alpha_j^*$ terms. This approximation is valid when $\omega_{mi} \gg \Omega_{12}, \Omega_{23}$.

In this case, we remind the reader that the current setup is in the classical regime. In the revised manuscript, we also pointed out that the Hamiltonian for the classical models can be quantized straightforwardly by quantizing α_i to the annihilation operator a_i and α_i^* to the creation operator a_i^\dagger , if the resonators work in the quantum regime.

Meanwhile, we want to point out that we have estimated the phonon number to be ~ 24 in current experimental setup. Although the resonators in the current setup should be treated as classical system, it is promising to approach the quantum regime in near future. This is discussed in the discussion section.

3. I still think that the authors are still excessively using dashed/dotted lines to indicate trends in their data. The data should speak for itself. Figure 2b is an example of this even in the revised manuscript.

Reply:

Thanks for the comment. We have deleted all the dashed/dotted lines in the figures of Figure 1 and Figure 2.

Reviewer #3 (Remarks to the Author):

This paper reports on the experimental demonstration of a tunable, Raman-type coupling of two separated mechanical modes in a graphene-based electromechanical system. As already mentioned in my previous report, I consider the paper very-well written and all the claims and conclusions are convincingly supported by experimental data and a theoretical model. In the resubmitted version the author have addressed all the points raised by the other referees, which further improved the quality of the manuscript. Although I still do not consider the demonstrated coupling as a major breakthrough, it is of relevance in the field of electromechanical systems and potentially in the future for the control of (quantum) mechanical transducers. Therefore, I recommend the paper for publication in Nature Communications.

Reply:

We thank the reviewer for his/her supportive comment and the recommendation of publication in the Nature Communications.

REVIEWERS' COMMENTS:

Reviewer #1 (Remarks to the Author):

I am satisfied with the responses from the authors and the revisions to the manuscript. I recommend its publication in Nature Communications.

Reviewer #2 (Remarks to the Author):

The authors have significantly improved the manuscript and clarified the aspects I had pointed out in the previous round of review. I recommend publication of this manuscript in Nature communications.

REVIEWERS' COMMENTS:

Reviewer #1 (Remarks to the Author):

I am satisfied with the responses from the authors and the revisions to the manuscript. I recommend its publication in Nature Communications.

Reply:

We thank the reviewer very much.

Reviewer #2 (Remarks to the Author):

The authors have significantly improved the manuscript and clarified the aspects I had pointed out in the previous round of review. I recommend publication of this manuscript in Nature communications.

Reply:

We thank the reviewer very much.